# Chronic Effects of Asymmetric and Symmetric Sport Load in Varsity Athletes across a Six Month Sport Season

**DOI:** 10.3390/ijerph20032186

**Published:** 2023-01-25

**Authors:** Valerio Bonavolontà, Maria Chiara Gallotta, Giovanna Zimatore, Davide Curzi, Dafne Ferrari, Maria Giulia Vinciguerra, Laura Guidetti, Carlo Baldari

**Affiliations:** 1Department of Biotechnological and Applied Clinical Sciences, University of L’Aquila, 67100 L’Aquila, Italy; 2Department of Physiology and Pharmacology “Vittorio Erspamer”, Sapienza University of Rome, 00185 Rome, Italy; 3Department of Theoretical and Applied Sciences, eCampus University, 22060 Novedrate, Italy; 4Department Unicusano, University Niccolò Cusano, 00166 Rome, Italy

**Keywords:** asymmetric load, spine, scoliosis, asymmetric and symmetric sports

## Abstract

The relation between specific sport practice and possible spine modifications is unclear. The aim of this study was to investigate the effects of different sports on the spine in adult varsity athletes across a six month sports season. Forty-four athletes (24.5 ± 3 years) were divided into two groups according to the typology of the sport practiced: the symmetric sports group (S, 22 athletes: track and field running, *n* = 14; cycling, *n* = 8), and the asymmetric sports group (A, 22 athletes: tennis, *n* = 22). The participants’ spines were evaluated with Formetric^®^ 4D rasterstereographic analysis at the beginning (BL), in the middle (INT), and at the end (FIN) of the season. Twenty-five parameters were measured in an average 4D modality. The results showed that the intervention factor (BL vs. FIN) had a significant effect on dimple distance (*p* < 0.05) and on left lateral deviation (BL vs. FIN and INT vs. FIN, *p* < 0.01 and *p* < 0.01, respectively). Statistical differences were found for the sport typology factor for pelvic antero-retroversion and right lateral deviation. For left lateral deviation, no modulation was found for the sport typology. Asymmetric versus symmetric sport loads showed small statistical differences in a non-professional sample of adult athletes. The practice of asymmetric sports should also be encouraged without exceeding the total number of hours per week.

## 1. Introduction

Sports disciplines are commonly divided into symmetric ones like gymnastics and running and asymmetric ones like tennis, fencing, and javelin throwing, despite the lack of a clear categorization [1,2,3,4].

Although several studies have highlighted this topic, there is still debate on the role that specific sports practices could exert on the postural and musculoskeletal systems.

Zaina [3] stated that the case of asymmetric sports is paradigmatic, as they are traditionally considered to favor musculoskeletal disorders and imbalances, without scientific evidence. The issue of functional asymmetries in the muscular system, such as the lower limbs or trunk and back muscles, has also been addressed with reference to several sports [5,6,7]. In a recent brief review, Maloney [8] reported that the asymmetric demands of sports will almost certainly result in asymmetric adaptations and proposed a new type of functional asymmetry called ‘sporting asymmetry’ that has to be linked to athletes’ long-standing participation in their sport. Conversely, a recent study reported that bilateral asymmetry could be normal in sport practice and not necessarily linked to decreases in performance or injuries [9], while previous evidence stated that young athletes may have spinal deformities that may be present per se or may be potentially associated with the practiced sport [10,11].

Moreover, the relationship between the asymmetric load and possible spine modifications or muscle asymmetry is still not clear, although several studies have been carried out [3,7,12,13,14,15].

Regarding spinal loading and sports practice, Wojtys et al. [16] reported that a significant increase in spinal curvature was associated with cumulative training exposure in adolescent athletes: gymnasts were the most affected athletes, followed by American footballers, hockey players, swimmers, and wrestlers. Bussey [6] concluded that athletes who begin participating in their unilateral sport pre-puberty may be at greater risk for asymmetrical pelvic adaptation, but it is yet to be established if the asymmetry led to the back pain or vice versa, as previously investigated [17]. In addition, in a long-term follow-up study comprising wrestlers, male gymnasts, soccer players, tennis players, female gymnasts, and a group of male non-athletes, significant thoraco-lumbar abnormalities were present only in the wrestler group [18].

Other studies suggested that training for different sports can generate significant asymmetry in the trunk muscles’ activation, particularly in the flexors [13], and that not all asymmetrical sports have been found to lead to significant paraspinal muscle asymmetry [6]. Zemkova et al. [19] found different ratios in golfers, ice hockey players, and tennis players with respect to their dominant/non-dominant sides. Sanchis-Moysi et al. found muscular asymmetry in professional tennis players, compared to non-active men and soccer players, respectively [14,15]. Additionally, in a recent study, Connolly et al. [20] reported that, although asymptomatic, previous magnetic resonance imaging spine analysis in tennis players has shown a good tendency to elevate the prevalence of radiological abnormalities mostly in the lumbar region.

Conversely, Zaina et al. [3] concluded in their cross-sectional study that the correlation between the most popular asymmetric sport, tennis, and spinal deformities in competitive adolescent players was not confirmed by their findings. In another cross-sectional study, Watanabe et al. [21] reported that in adolescent students, only those with previous classical ballet training had showed higher odds of developing Adolescent Idiopathic Scoliosis (AIS), whereas students playing asymmetric sports such as basketball and badminton did not show the same relation. In line with the previous study, Zaina et al. [3] supported the idea that a massive amount of practice per week could have an influence on spinal alterations rather than the typology of sport. On this basis, Jayanthy [22] reported that adolescent athletes may be at risk if they spend “more hours per week in sports than years they are old” and are potentially exposed to any injury.

Thus, even if it has been hypothesized that asymmetric load, over several years of practice, could induce chronic modifications on the spine of expert athletes, this argument is not confirmed by the research conducted, which showed sharply contrasting results.

Moreover, to our knowledge, no studies have investigated the chronic effects of different sports disciplines on the spine over a period of several months or examined a non-professional population. Our previous study investigated the acute effects of asymmetric load [23] in two different single asymmetric sessions and the possible morphological alterations in recreational players. The modifications induced were linked to the total workload of the training session and to the expertise of the subjects. Then it is hypothesized that a long-lasting asymmetric practice could induce a different effect on the spine with respect to the symmetric one, as it has been verified in acute in expert athletes [23].

Therefore, the aim of this study was to evaluate the effects of a 6 month sport practice on the variables related to the dorso-lumbar spine in two groups of varsity athletes, one practicing symmetric sports and the other asymmetric sports.

## 2. Materials and Methods

### 2.1. Participants

A priori analysis with G*Power indicated that 44 subjects were sufficient to detect a medium effect size (f = 0.25) with a coefficient of correlation of r = 0.5, 95% power, and an α = 0.05 using a within-between subjects design. Therefore, 44 male varsity athletes were recruited from the sport center of the university and were divided into two groups according to the typology of the sport team they belonged to (Table 1). The first group (S) was formed by 22 subjects practicing a symmetric discipline, in particular track and field runners (*n* = 14) and cyclists (*n* = 8). The second group was formed by 22 subjects practicing an asymmetric discipline (tennis players, *n* = 22).

The study inclusion criteria were:-Healthy skeletal system;-Eligibility for medical certification for competitive sports;-Age older than 19 years;-Right-handed laterality in order to isolate this variable;-At least 6 years of practice in the respective sport;-A participation in the national university championships.

Exclusion criteria were:-Profiles of elite or high-level athletes from the past or present;-Skeletal or muscular pathologies.

All athletes were monitored during their usual training under the supervision of the university coaches and had a similar lifestyle.

The study was a non-interventional/observational study based on the definitions of the European Directive 2001/20/EC, for which the approval of an ethics committee was not requested (The European Parliament and the Council, 2001). However, an Institutional Review Board (approval code 70/11) has approved the study, which was conducted according to the Declaration of Helsinki and followed the International Code of Ethics for Occupational Health Professionals (International Committee of Occupational Health, 2014). All participants gave their written informed consent before participating in the study.

### 2.2. Procedures

Participants underwent three different spine evaluations, respectively, at the beginning, in the middle, and at the end of the academic year, which roughly corresponds to three significant moments in the university sport center schedule plan: the initial evaluation was conducted in November, when the sport center usually ends the initial recruiting selection and training activities start; the intermediate evaluation was conducted in February, after the first half of training sessions; and the final evaluation was conducted in May, when they had concluded the second half of training sessions and before the participation at the national university championships. An assessment was carried out in the morning in laboratory conditions by the same expert operator, who had also followed the manufacturer training course and who conducted all the evaluations for both groups. Subjects were evaluated without shoes and wearing only underwear under the bottom. Subjects were positioned on a hard, flat, even surface, and the position was standardized and kept for the three measurements across the season.

### 2.3. General Description of the Training of the Two Groups

Both groups trained with their respective sports teams with an average frequency of three times per week; each single session lasted between one hour and a half and two hours. In order to evaluate the contribution of the different sports disciplines on athletes’ spines, a pre-training meeting was held with the coaches responsible for each sports team. The standardization of the training schedule plan and the modulation of training parameters (volume and intensity) were carried out. As the participants all belonged to the same university sport center, the three different disciplines considered (track and field running, cycling, and tennis) shared the same periodization and scheduling of the sport season as follows: in the first part of the year, the training program was focused on the general physical preparation and on the sport discipline’s motor skills enhancement (off-season preparation period); in the second part, it was focused on the technical skills’ improvement (pre-competition period); and in the third part, it was focused on the preparation of the competitions (in-season period), i.e., the participation in the national university championships. At the end of each training period, the corresponding spine evaluation was performed, namely baseline (BL), intermediate (INT), and final (FIN), respectively.

### 2.4. Instrumentation

Athletes’ dorsal and lumbar spines were evaluated using back surface rasterstereographic analysis with the Formetric 4D system (Diers International GmbH, Schlangenbad, Germany), which allows for a tridimensional reconstruction of the dorso-lumbar spine starting from the surface analysis of the back [24].

Previous studies had shown from good to excellent reliability for this device [24,25] and good accuracy compared to X-rays, the gold standard [26]. For the present study, an average 4D modality was used, which provides an averaging measurement based on 12 subsequent images in a recording interval lasting about 6 s. The following 25 parameters were measured and taken into consideration: trunk length (mm), dimples distance (mm), antero-posterior flexion VPDM detected from vertebra prominens (VP, corresponding to C7 vertebra) to midpoint of lumbar dimples (DM) (°), antero-posterior flexion VPDM (mm), lateral flexion VPDM (°), lateral flexion VPDM (mm), pelvic inclination (°), pelvic inclination (mm), pelvic torsion (°), pelvic inclination (dimples) (°), pelvic rotation (°), kyphotic apex (mm), inflection point ITL (mm), lordotic apex (mm), inflection point ILS (mm), cervical fleche (mm), lumbar fleche (mm), kyphotic angle (ICT-ITL) (°), lordotic angle (ITL-ILS) (°), pelvic antero-retroversion (°), surface rotation (right) (°), surface rotation (left) (°), trunk torsion (°), lateral deviation VPDM (right) (mm), and lateral deviation VPDM (left) (mm). These parameters are explained in detail in Table 2.

### 2.5. Statistical Analysis

The mean scores and standard deviations (m ± s) for BL, INT, and FIN intervention evaluations were calculated separately for the symmetric (S) and asymmetric (A) groups. Between the groups, differences in the baseline spinal variable scores were verified by means of an unpaired comparison *t*-test. Thus, analysis of variance (ANOVA) for repeated measurements was applied to compare spine alteration changes (the intervention effect), accounting for sport typology (S vs. A) and intervention (BL vs. INT vs. FIN). Thus, for all variables a 2 × 3 mixed-model analysis of variance (ANOVA) for repeated measures with the between-group factor sport typology (S vs. A) and within-subjects factor intervention (BL vs. INT vs. FIN) was performed. The effect size was also calculated using Cohen’s definition of small, medium, and large effect sizes as partial ƞ2 = 0.01, 0.06, and 0.14, respectively [27]. Significant main effects or interactions were further analyzed by means of the Bonferroni post hoc analysis. A statistical significance was set as *p* ≤ 0.05. The statistical package SPSS (SPSS Inc., Chicago, IL, USA) version 24.0 for Windows was used for all statistical analyses.

## 3. Results

The mean values and SD for BL, INT, and FIN intervention spinal assessments of S and A athletes are shown in Table 3.

Differences in the BL spine assessment scores of groups S and A were verified (*p* ≤ 0.05), and significant differences were revealed for right lateral deviation and pelvic antero-retroversion. Specifically, for right lateral deviation, mean values and standard deviations at BL were 10.43 ± 5.68 mm and 5.93 ± 3.81 mm (*p* < 0.01) for the A and S groups, respectively, while for pelvic antero-retroversion, mean values and standard deviations were 20.77 ± 7.60° and 16.27 ± 7.32° (*p* < 0.05) for the A and S groups, respectively.

ANOVA revealed a significant main effect of intervention on dimple distance (F_2,540_ = 4.19, *p* = 0.01, and ƞ2 = 0.091) and on left lateral deviation VPDM (F_2,219_ = 6.63, *p* = 0.002, and ƞ2 = 0.136). Specifically, athletes significantly increased their dimple distance after intervention (Figure 1, panel “a”) and significantly decreased their left lateral deviation VPDM in INT evaluation (Figure 1, panel “b”). Moreover, ANOVA revealed a significant main effect of sport typology on pelvic antero-retroversion (F_1,2202_ = 4.75, *p* < 0.05, and ƞ2 = 0.102) and lateral deviation VPDM (right) (F_1,788_ = 9.84, *p* = 0.003, and ƞ2 = 0.190). Specifically, athletes of group A showed higher pelvic antero-retroversion values (20.80 ± 7.24° vs. 16.04 ± 7.47°, *p* < 0.05, respectively) and higher lateral deviation VPDM (right) values (10.54 ± 5.68 mm vs. 6.44 ± 3.58 mm, *p* = 0.003, respectively) with respect to athletes of group S.

## 4. Discussion

This study aimed to evaluate the dorso-lumbar spine of two groups (S and A) of adult varsity athletes, respectively practicing symmetric and asymmetric sports, three times across a 6 month season through rastersterographic analysis. The current paper is a natural extension of a previous study conducted by the authors in acute [23] over a mid-term period, i.e., in chronic. Previous studies that reported an increase in the incidence of scoliosis in asymmetrical disciplines athletes but were not clinically relevant [3,28,29] led to the expectation of finding small and not alarmingly significant differences between the groups.

Indeed, results showed small statistical differences for the factor sport typology for two of the twenty-five parameters measured. Pelvic antero-retroversion and right lateral deviation were found to be higher in the A group, also before intervention, possibly because all the subjects belonging to this group were right-handed and the years of practice could have led to this statistical but not clinical difference; moreover, both parameters are not linked to rotation on the transverse plane as they are, respectively, referred to the sagittal and frontal planes, thus not having incidence on a possible tridimensional spine modification.

Furthermore, while the subjects in this study had previously competed in university national championships, and the majority of them had also competed in individual competitions in their respective disciplines, they were not elite or high-level athletes. Therefore, due to the sample characterization, the load induced by the sport discipline may have produced ineffective changes. Thus, it could be argued that asymmetric sports, such as tennis, are not able to induce significant modifications in the main postural parameters if not practiced in a large and professional setting. These results are in line with Zaina et al. [3], who reported that a tennis practice of up to 5 h a week is not likely to induce pathological alterations or even negative postural effects such as low back pain. Moreover, some studies that were previously conducted on samples from professional or high-level athletes were not clinically alarming either [28,29,30]. This appears to be a fundamental signal that can reassure parents and clinicians about children’s and adolescents’ spine health, and that can stop claiming the potential negative role of practicing amatorially or even at a competitive level asymmetric sports like tennis, fencing, etc., when the weekly amount does not exceed 4–5 h, as it happens in most cases both during young stages and adulthood.

Instead, referring to the intervention effect, results showed statistical significance for the parameters dimples distance (BL vs. FIN, *p* < 0.05) and left lateral deviation (BL vs. INT, *p* < 0.01; INT vs. FIN, *p* < 0.01). Dimples distance is an invariant parameter, i.e., it is not dependent on the subject’s positioning in relation to the machine; it could be hypothesized that a six month sport season would result in an increase in this parameter in all participants due to the pelvic mass hypertrophy process. With reference to left lateral deviation, no modulation was found for sport typology, but for all athletes, intervention had an increasing effect, suggesting that a deviation on the frontal plane can occur independently by the symmetric/asymmetric load especially in the pre-competition period (INT), which has shown statistical differences in respect to the BL and FIN; in fact, in this period, training is more focused on technical elements that are hypothetically less stressing for the spine, though INT values showed a decrease trend for all participants. However, these two parameters showed statistical variations much lower, respectively, than the between-day smallest detectable changes (SDC) reported by Degenhardt et al. [31].

Thus, the present findings support the idea that 6 months of asymmetric/symmetric sport practice do not lead to significant differences in young adult athletes’ spines. It must also be considered that, although not elite or high-level athletes, all subjects involved in the study were healthy and trained students from a physical education university course, thus likely representing an above-average sample.

On the other hand, previous research [5,32] suggested that balance and core training and compensative exercises are capable of counteracting/reducing the degree of asymmetry in lower-limb strength in young tennis players, as well as strength and postural control training in other sport disciplines [33]. In addition, as the sport-vision area has become widespread in athletes’ training and in the general population, oculo-motor training can represent another way to improve some postural parameters [34]. Although certain muscular asymmetries and imbalances must be functionally attributed to the specificity of some sport disciplines [8], modern training must consider the importance of a harmonic and holistic approach to physical and athletic development in order to optimize performance and decrease sporting asymmetries and thus injury risk. In line with this, in a 15 year follow-up study, Baranto et al. [35] reported that, in four different top athletes sports, most of the degenerative abnormalities, probably due to a combination of high-load and ageing, were present also at baseline (i.e., 15 years before the follow-up), and thus, it is crucial to adopt preventive measures to avoid the development of back injuries in young athletes.

Nonetheless, it has been suggested that double-major adolescent idiopathic curves were more related to physical and sport activities practice, especially gymnastics, than the single-major curve, implying a potential beneficial compensatory effect of sport practice [36]. Indeed, in a historical longitudinal study, it has also been proposed that increased physical activities can be considered as a possible therapy that may protect against AIS by involving neuromuscular feedback mechanisms common to all joints [37]. It could also be hypothesized that training programs should reduce interlimb asymmetry while increasing performance, as has previously been reported [38] (Bishop, 2018).

Therefore, the general finding of the present study, in line with the above-mentioned studies, suggests that therapists and physicians should encourage their patients to regularly and non-intensively practice physical and sport activities to help increase balance, proprioception, and sensorimotor levels without discouraging *a priori* the practice of asymmetrical sports.

The present study has some limitations, including the non-elite sport level of the sample, the duration of the observation period, which lasted six months, and the number of evaluations performed, which was limited to three. We did not perform a diagnostic evaluation because of the healthy status of our sample. However, rasterstereography offers a non-invasive, low-cost, and X-ray-free evaluation that has good accuracy and reliability [24,31].

## 5. Conclusions

In line with previous studies, the present findings support the idea that the practice of asymmetric sports does not per se represent a danger for the dorso-lumbar spine of healthy young adults who train for college competitions, especially when this practice does not exceed a weekly amount of 4–5 h. Therefore, and in accordance to the specific existing literature, practitioners and sport educators could play a role in reassuring pupils and their families that it is tendentially safe to practice asymmetric disciplines, albeit not excessively, without negative consequences on musculoskeletal development. However, compensatory and harmonic sport training is advisable to optimize performance and reduce non-contact injuries and stress overloading in general, and especially for asymmetrical sports disciplines. 

## Figures and Tables

**Figure 1 ijerph-20-02186-f001:**
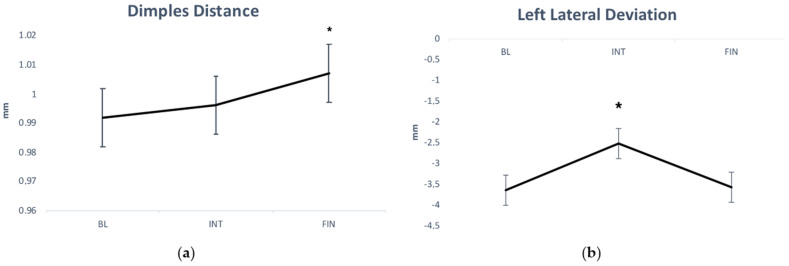
Variation in the dimples distance (**a**) and left lateral deviation (**b**) parameters across the three measurements. BL: baseline. INT: intermediate. FIN: final. *: *p* < 0.05 BL vs. FIN (panel a); *p* < 0.05 INT vs. BL and vs. FIN (panel b).

**Table 1 ijerph-20-02186-t001:** Sample description. The data are expressed as mean ± SD.

Group	N	Age	Height	Weight
Symmetric (S)	22	25 ± 3.1 years	168.2 ± 38.1 cm	67.4 ± 17.4 kg
Asymmetric (A)	22	24 ± 2.9 years	176.4 ± 8 cm	72 ± 10.8 kg
*t* test (*p*-value)		0.07	0.92	0.65

N = number of subjects.

**Table 2 ijerph-20-02186-t002:** Definitions of the spinal parameters measured by the Formetric 4D system.

Parameter	Unit	Description
Trunk length	mm	The distance from the vertebra prominens (VP) to the midpoint of the lumbar dimples (DM)
Dimples distance	mm	The distance from dimple left (DL) to dimple right (DR)
Antero-posterior flexion VPDM (trunk inclination)	°	The angle between the line connecting VP-DM and an external plumb line
Antero-posterior flexion VPDM (trunk inclination)	mm	The distance between VP and the connecting external plumb line
Lateral flexion VPDM (trunk imbalance)	°	The angle between the line connecting VP-DM and a plumb line passing through VP
Lateral flexion VPDM (trunk imbalance)	mm	The lateral distance between VP and DM
Pelvic inclination	°	The angle between the line connecting DL and DR and the horizontal
Pelvic inclination	mm	The difference in height between DL and DR
Pelvic inclination (dimples)	°	The mean vertical components of the surface normals at DL and DR
Pelvic torsion	°	The torsion of the surface normal of the DL and DR
Pelvic rotation	°	In the frontal plane, the angle of rotation of DR in relation to DL
Kyphotic apex	mm	The location of the posterior apex of the sagittal profile
Inflection point ITL	mm	The point of maximum negative surface inclination between the kyphotic apex (KA) and the lordotic apex
Lordotic apex	mm	The location of the frontal apex of the sagittal profile in the lower region
Inflection point ILS	mm	The point of maximum positive surface inclination in the region between the lordotic apex (LA) and the sacrum
Cervical fleche	mm	The horizontal distance between the cervical apex and the tangent through the KA
Lumbar fleche	mm	The horizontal distance between the LA and the tangent through the KA
Kyphotic angle ICT-ITL	°	The angle between the surface tangents and the ICT and ITL
Lordotic angle ITL-ILS	°	The angle between the surface tangents from ITL and ILS
Pelvic antero-retroversion	°	The angle of the vertical surface normals from the horizontal of the DM
Right surface rotation	°	The maximum value of the horizontal components of the surface normals on the symmetry line to the right
Left Surface rotation	°	The maximum value of the horizontal components of the surface normals on the symmetry line to the left
Trunk torsion	°	The maximum value of the horizontal components on vertebra prominens compared to the horizontal components of the symmetry
Right Lateral deviation VPDM	mm	The maximum deviation of the midline of the spine from the VP-DM line to the right
Left Lateral deviation VPDM	mm	The maximum deviation of the midline of the spine from the VP-DM line to the left

Adapted from the DIERS Formetric III 4D manual. DL: sacral dimple left; DR: sacral dimple right; ICT: cervicothoracic transition point; ILS: lumbosacral transition point; ITL: thoracolumbar transition point; VP: vertebral prominens; DM: midpoint of lumbar dimples; KA: kyphotic apex; LA: lordotic apex.

**Table 3 ijerph-20-02186-t003:** Baseline-, intermediate-, and final-intervention spine assessment (mean values ± s) of S and A groups.

Group	S Group			A Group		
Evaluation	Baseline	Intermediate	Final	Baseline	Intermediate	Final
Parameter						
Trun length (mm)	469.4 ± 22.8	470.4 ± 24.6	471.1 ± 23.1	468.6 ± 30.9	472.6 ± 24.6	470.2 ± 30.4
Dimples distance (mm)	97.5 ± 9.3	98.9 ± 9.2	100.1 ± 11.8	100.8 ± 10	100.3 ± 10.3	101.2 ± 10.2
Antero-posterior flexion VPDM (°)	3.1 ± 2.2	3.1 ± 2.4	2.8 ± 2.4	3.1 ± 2.4	3.1 ± 2.7	3.2 ± 2.5
Antero-posterior flexion VPDM (mm)	24.3 ± 18.6	23.3 ± 21.2	21.4 ± 19.7	25.4 ± 19.4	25.8 ± 22.6	25.1 ± 20.1
Lateral flexion VPDM (°)	−0.7 ± 1.2	−1 ± 1.1	−0.9 ± 1.2	−0.7 ± 0.9	−0.9 ± 0.8	−0.6 ± 0.9
Lateral flexion VPDM (mm)	−5.6 ± 9.4	−7. ± 7.5	−5.5 ± 8.8	−6 ± 7.1	−7.1 ± 6.9	−7.1 ± 7.6
Pelvic inclination (°)	−1 ± 3.4	0.3 ± 4	0.2 ± 3.1	−1.5 ± 2.9	−1.1 ± 3	−1 ± 3.2
Pelvic inclination (mm)	−1.7 ± 5.8	0.2 ± 7.3	0.9 ± 5.4	−2.6 ± 5.1	−1.7 ± 5.0	−2.3 ± 4.9
Pelvic torsion (°)	1.5 ± 2.6	1 ± 3.6	1.1 ± 2.5	1.3 ± 2.8	0.4 ± 3.3	1.4 ± 2.9
Pelvic inclination (dimples) (°)	18.4 ± 6.5	18.9 ± 6.8	19.0 ± 6.8	21.4 ± 5.0	21.2 ± 4.3	21.5 ± 4.5
Pelvic rotation (°)	0.15 ± 3.1	−0.6 ± 3.1	−2.6 ± 7.9	−0.8 ± 3.4	−6.0 ± 3.1	−1.1 ± 3.8
iphotic apex (mm)	−183.3 ± 16.6	−182.6 ± 19.6	−182.5 ± 21.3	−181.4 ± 25.2	−184.0 ± 26.7	−181.7 ± 26.2
Inflection point ITL (mm)	−306.4 ± 42.1	−311.7 ± 42	−308.3 ± 44.0	−303.3 ± −300.6	−300.6 ± 38.2	−300.6 ± 38.2
Lordotic apex (mm)	−395.5 ± 28	−395.5 ± 27.2	−397.8 ± 30.6	−390.7 ± 31.9	−393.5 ± 29	−391.0 ± 30.9
Inflection point ILS (mm)	−480.6 ± 30.6	−482.4 ± 30.2	−485.4 ± 28.8	−470.74 ± 35.6	−473.7 ± 32.3	−457.3 ± 88
Cervical fleche (mm)	73.4 ± 13.6	71.8 ± 16	72.2 ± 15.5	71 ± 15.5	70 ± 17.1	71.8 ± 16.2
Lumbar fleche (mm)	38.7 ± 11.7	39.6 ± 12	41 ± 11.3	43.2 ± 13.7	44.1 ± 12	43.7 ± 12.1
iphotic angle (ICT-ITL max) (°)	49.4 ± 8.5	49.2 ± 6.9	50.5 ± 8.2	49 ± 6.8	50.7 ± 8.7	48.6 ± 7.9
Lordotic angle (ITL-ILS max) (°)	37.6 ± 8.1	38.1 ± 6.6	39.3 ± 11.9	40.3 ± 9.1	40.9 ± 9.0	40.3 ± 9.5
Pelvic antero-retroversion (°)	16.3 ± 7.3	15.7 ± 7.3	16.1 ± 8	20.8 ± 7.6	20.7 ± 7	20.9 ± 7.4
Surface rotation (right) (°)	3.4 ± 3.2	2.9 ± 2.3	3.7 ± 3.4	2.7 ± 3	2.3 ± 2.4	2.4 ± 3.0
Surface rotation (left) (°)	−4.7 ± 2.5	−4.7 ± 2.6	−6.5 ± 3.8	−6.2 ± 3.4	−6.2 ± 3.3	−4.8 ± 2.3
Trun torsion (°)	3.1 ± 6.5	−0.9 ± 6.2	1.3 ± 5.3	−0.4 ± 5	−0.8 ± 5.4	−0.2 ± 4.1
Lateral deviation VPDM R (mm)	10.4 ± 5.7	10.4 ± 6.2	10.8 ± 5.4	5.9 ± 3.8	6.6 ± 3	6.8 ± 4
Lateral deviation VPDM L (mm)	−4.6 ± 4.4	−3.1 ± 3.6	−4.4 ± 4.1	−2.7 ± 3	−1.9 ± 2.2	−2.7 ± 2.7

VPDM: from vertebral prominens (VP) to midpoint of lumbar dimples (DM); ICT: cervicothoracic transition point; ILS: lumbosacral transition point; ITL: thoracolumbar transition point; R: right. L: left.

## Data Availability

The dataset will be made available upon reasonable request.

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
