# Peer review of "Chronic Effects of Asymmetric and Symmetric Sport Load in Varsity Athletes across a Six Month Sport Season"

_ijerph, 2023, doi:10.3390/ijerph20032186_

Round 1

Reviewer 1 Report

Dear Authors,

It is my pleasure to review your study but I have a few doubts.

General information:

-abstract should be prepare in accordance with the guidelines  (divided into sections),

-references should be newer, please correct it.

Introduction:

-introduction is a bit too long.

-the aim of the work should be clarified, please correct it.

-no hypothesis, this should be corrected.

M&M:

-were the two study groups different from each other? Please provide calculations.

-whether the sample size of this group was counted?

-the description of the studied groups is difficult to read, it is better to present it in a table.

-please describe in more detail the inclusion and exclusion criteria.

- in 2.2. Procedures - what was the assessment? Who conducted it? How? Lack of information. It should be corrected.

-"These parameters as well as the rasterstereographic analysis methodology are explained in detail in previous works by the authors [22, 24]."

It doesn't matter. This should be discussed in this paper. What is the risk of measurement error? 

Results:

-the abbreviations in table 1 should be explained,

-table 1 is difficult to interpret, there is no precise description of the value, it should be corrected,

Discussion:

-limitations of study should be added. 

Conclusion:

-Has a medical examination been carried out? Has imaging of the spine been performed? Such conclusions cannot be made without proper medical research. And the device for evaluating the parameters that are described is not a diagnostic medical standard. Conclusions need to be improved.

Reviewer 2 Report

The distribution of both groups by gender is not shown. A single presentation of a few of the examined features is not sufficient to draw the presented conclusions. Asymmetric movement patterns give increasing effects after the age of thirty and later, they are associated with the long-term effect of gravity on asymmetrically fixed structures. In addition, we do not know the functional state of the musculoskeletal system of the subjects, e.g. muscle tension, soft tissue pain, functional tests, e.g. for sacral joints. A symmetrical sport practiced simultaneously with an asymmetrical one does not compensate for the asymmetry. In asymmetric sports, special individual training should be conducted to compensate for postural and functional asymmetries, especially in young players. Prophylactically, posture correction should be recommended for all athletes.

Round 2

Reviewer 1 Report

Dear Authors,

thank you for the changes made. The manuscript looks much better.

The inclusion and exclusion criteria could be better presented.

I suggest you write these criteria from hyphens.

It will be better for the reader. 

Besides, the authors corrected the manuscript well.

Best regards.

Author Response

Dear Reviewer,

thank you again for this suggestion.

Inclusion and exclusion criteria were modified accordingly.

Best Regards